# Assessment of the Negative Impact of Urban Air Pollution on Population Health Using Machine Learning Method

**DOI:** 10.3390/ijerph20186770

**Published:** 2023-09-15

**Authors:** Nurlan Temirbekov, Marzhan Temirbekova, Dinara Tamabay, Syrym Kasenov, Seilkhan Askarov, Zulfiya Tukenova

**Affiliations:** 1National Engineering Academy of RK, Almaty 050010, Kazakhstan; temirbekov@rambler.ru (N.T.); dtamabay@gmail.com (D.T.); syrym.kasenov@gmail.com (S.K.); 2Faculty of Mechanics and Mathematics, Al-Farabi Kazakh National University, Almaty 050040, Kazakhstan; 3Almaty University of Power Engineering and Telecommunications Named after G. Daukeyev, Almaty 050013, Kazakhstan; 4Ecoservice-S Limited Liability Partnership, Almaty 050009, Kazakhstan; askarov.s@ecoservice.kz; 5Institute of Zoology of the Ministry of Higher Education and Science of the RK, Almaty 050060, Kazakhstan; otdel_nauki8@mail.ru

**Keywords:** respiratory diseases, air pollution, machine learning algorithms, random forest, recommendations

## Abstract

This study focuses on assessing the level of morbidity among the population of Almaty, Kazakhstan, and investigating its connection with atmospheric air pollution using machine learning algorithms. The use of these algorithms is aimed at analyzing the relationship between air pollution levels and the state of public health, as well as the correlations between COVID-19 infection and the development of respiratory diseases. This study analyzes the respiratory diseases of the population of Almaty and the level of air pollution as a result of suspended particles for the period of 2017–2022. The study includes recommendations to reduce harmful emissions into the atmosphere using machine learning methods. The results of the study show that air pollution is a critical factor affecting the increase in the number of diseases of the respiratory system. The study recommends taking measures to reduce air pollution and improve air quality in order to prevent the development of chronic respiratory diseases. The study offers recommendations to industrial enterprises, traffic management organizations, thermal power plants, the Department of Environmental Protection, and local executive bodies in order to reduce respiratory diseases among the population.

## 1. Introduction

The increase in the incidence of respiratory diseases (RD), particularly chronic bronchitis, chronic obstructive pulmonary disease (COPD), and asthma, may be associated with a high degree of atmospheric air pollution. Every year, 3 million people in the world die as a result of diseases caused by environmental pollution, including 1.7 million children under the age of 5 (according to WHO, 2017) [1].

Long-term exposure to air pollution will impact the human immune system, making it more susceptible to various types of respiratory diseases. It is evident that COVID-19 is a respiratory illness. It has been demonstrated in [2] that COVID-19 is detected in aerosols for up to three hours, and air pollution creates favorable conditions for the virus’s spread [3]. According to data from [4,5], the number of COVID-19 cases in Almaty in the period from 2020 to the present year was 259,279, and the death rate is 2625 people, which is 20% of all deaths in Kazakhstan.

In work [6], an epidemiological analysis with an emphasis on RD among the population of Almaty in the period from 2013 to 2017 was conducted. In this study, the correlation between the prevalence of RD and the levels of suspended solids in the air over the same period was studied, and for a retrospective study, information on the concentration of PM_2.5_ in the city’s atmosphere was obtained using generally accepted statistical methods. The standard deviation was calculated for numerical data, the level of statistical significance was calculated using the generally accepted Student’s *t*-test, and the Pearson correlation coefficient was determined (using Microsoft Excel). This type of research can be carried out using machine learning (ML) methods, rather than conventional statistical methods.

Air pollution leads to the disruption of ecosystems and significant economic and social damage, and it also affects the health of the population. In 2020, Kazakhstan ranked 32nd in the world in terms of pollution, with an average annual PM_2.5_ concentration of 21.9 μg/m^3^. Exceeding the concentration of PM_2.5_ annually led to 8134 cases of early death of adults in 2015–2017 in 21 cities of Kazakhstan [7].

Almaty is affected by temperature inversion due to its geographical location and the characteristics of the air flow because of its mountainous terrain. This inversion makes it much more difficult for suspended particles and pollutants to enter atmospheric air. Currently, Almaty uses three combined thermal power plants (CHP-1, CHP-2, and CHP-3) to generate electricity and heat. CHP-2 is located within the city, while CHP-3 is three kilometers away from it. While CHP-1 uses natural gas as fuel, the other two power plants burn about 3.7 million tons of low-grade coal annually with high ash contents (39–40%) [8]. These coal-fired power plants do not use advanced emission control technologies, such as desulfurization and denitrification, which leads to the improper removal of pollutants. As of May 2022, a total of 508.7 thousand passenger cars were registered in Almaty [9], and additional vehicles from suburban areas contribute to the volume of traffic [10]. However, the number of scientific studies assessing the air quality in Almaty is limited. In order to comprehensively address the problem of air pollution and plan to reduce pollutant emissions, it is necessary to develop recommendations for the operation of thermal power plants and transport, and legislative measures are also needed.

In addition to policy and legislative changes, the development of predictive models for PM_2.5_ (particulate matter with a diameter of 2.5 microns or less) is crucial for reducing air pollution. These models can provide the population with information in advance about periods of high pollution levels, allowing them to plan accordingly. Forecasting and assessing the state of PM_2.5_ particles have always been difficult tasks due to their temporal and spatial variations and their dependence on various atmospheric conditions. Various methods and statistical models were used to predict PM_2.5_ levels [11,12]. The authors of work [13] built an ensemble machine learning model based on daily data on PM_2.5_, visibility, and other meteorological data in order to predict its daily concentrations in regions with no observations conducted. Compared with traditional statistical models, the proposed machine learning methods, such as deep learning, gradient boosting, random Forest, and ensemble methods, improved the accuracy of predicting daily concentrations of PM_2.5_. However, statistical models have limitations and are less informative when dealing with non-linearity and data change over time compared to PM_2.5_ models based on artificial intelligence [14]. Machine-learning-based methods used in PM_2.5_ forecasting have shown different results in different geographical locations, such as Shanghai, Beijing, Tehran, and Ho Chi Minh City, due to temporal and spatial characteristics and weather conditions. Hourly and multi-stage PM_2.5_ forecasting, which is more complex, presents a bigger problem than average daily and single-stage PM_2.5_ forecasting for one day or hour. In Vietnam, recent research has focused on creating a predictive model to estimate average daily levels of PM_2.5_ in Ho Chi Minh City. These studies used data obtained from a single source located in the United States. However, the PM_2.5_ daily forecasting model does not provide sufficient time detail for people to plan their day in advance, especially during periods of high atmospheric pollution at different times during the day. Moreover, the forecasting model for a single location lacks the necessary informativeness to represent the entire city due to differences in PM_2.5_ concentrations in different locations, which are affected by different peaks in traffic intensity, industrial activity, and residential areas. Moreover, in article [15], models were developed to predict the average daily PM_2.5_ index in accordance with the National Technical Regulation of Vietnam on the Assessment of Ambient Air Quality in the Atmosphere QCVN 03:2015 (MONRE, 2013) [16] and the global guidelines for air quality management established by the World Health Organization (WHO, 2021) [17].

The expansion of the economic and industrial sectors poses a threat to the preservation of sustainable development. Sustainable development is aimed at promoting “green” practices that reduce carbon dioxide emissions throughout the life cycle of products, including production, transportation, use, and disposal. Some industrial emissions, especially suspended particles with a diameter of less than 2.5 microns (PM_2.5_), can be toxic and have an adverse effect on human health. The results of many studies show that an increase in the level of PM_2.5_ largely leads to RD [18] and the development of cancer [19].

The concentration of PM_2.5_ includes a wide range of chemical components, such as sulfates, nitrates, ammonium compounds, carbon-containing compounds, elements from the earth’s crust, organic compounds, and free-type radicals [20,21]. Natural sources include dust, soil, and biomass, while anthropogenic sources include emissions from vehicles, coal and gasoline burning, petrochemical production, and steel mills. Secondary sources can be formed as a result of the photochemical conversion of harmful substances such as SO_2_ (sulfur oxide) and NO_x_ (nitrogen oxides). The effects of air pollution caused by these factors can vary in scale and duration. The movement of vehicles usually leads to the contamination of the roadway. As soon as PM_2.5_ particles enter the atmosphere from any source, they disperse into the environment and accumulate in certain regions, especially in valleys, depressions, and hillsides. These particles can persist throughout the day and are carried by wind currents. If weather conditions prevent their dispersion, this accumulation can persist for a long period. Temperature inversion, characterized by the formation of a layer of warm air over a colder earth, is one such weather condition that can trap PM_2.5_ particles under a layer of warm air [15].

Climate conditions, weather, and air pollution have a significant impact on the respiratory organs of people and are associated with the occurrence of RD. Climatic conditions such as temperature, humidity, and seasonal changes can affect the frequency and spread of RD. For example, some studies show that low temperatures and dry air in the winter months contribute to an increase in the spread of influenza and other respiratory infections. Certain climatic conditions can also affect the level of allergens and pollen in the air, which can worsen symptoms in people with allergic RD. Air pollution is another factor that can negatively affect the respiratory system and increase the risk of developing RD. Air pollutants such as particulate matter, smoke, automobile emissions, and industrial waste can irritate the lungs, cause inflammation, and increase the risk of infections. Prolonged exposure to polluted air can increase the frequency and severity of RD, especially in people with pre-existing respiratory system problems [22,23]. ML models have the potential to use data on air pollution and other environmental factors to predict the occurrence of RD. ML algorithms can analyze large amounts of data and identify links between air pollution, climatic conditions, and diseases. This can help in the development of forecasting and warning systems, which will make it possible to respond more effectively to health threats and appropriate measures can be put in place to prevent and treat RD [24,25].

Scientists are developing various methods, including chemical equations and physical and mathematical modeling, to predict future levels of air quality. However, these models are not yet fully exploiting advances in ML and continue to rely on physics, mathematics, and statistics. Although these models offer advantages in solving non-stationary processes and automating calculations, they face limitations when processing large data sets. To overcome this, researchers turn to ML to predict air quality and its qualitative characteristics [26,27,28]. Monitoring systems equipped with measuring instruments collect extensive data sets on the concentrations of harmful substances in the air, which allows scientists to use various ML algorithms to predict future levels of air pollution [29]. ML methods have found application in many fields, with the first mentions in the field of environmental science dating back to the 1990s [30]. However, with the increasing availability of diverse data sets relating to various aspects of the planet, there is a growing interest in using ML methods to protect the environment. Unlike traditional statistical methods, ML has the advantage of allowing for a more comprehensive analysis and evaluation of data. This is explained by its ability to model complex and nonlinear relationships inherent in the natural environment [31].

Large particles PM_2.5_-PM_10_: These particles have aerodynamic characteristics and a diameter ranging from 2.5 to 10 microns. They are mainly deposited in the primary bronchus. Small particles PM_2.5_: This group includes particles ranging in size from 0.1 to 2.5 microns. These particles have the ability to penetrate through the alveoli, which are tiny air sacs in the lungs. Ultrafine PM_0.1_: These particles have a diameter of less than 0.1 microns. They are able to appear in cell membranes and interact directly with cells and various structures [32]. The study presented in [33] uses ML to estimate the concentration of PM_2.5_ in Thailand. The researchers used aerosol reanalysis data and NASA MERRA 2 meteorological data to prepare and validate a controlled random forest MLA (machine learning algorithm). The trained algorithm demonstrates an average deviation close to zero across the country and demonstrates its effectiveness in fixing the daily cycle at different times during one year. The aim of the study conducted in [34] was to assess visibility in Seoul, which is one of the most polluted cities in South Korea, using MLA based on meteorological data and suspended solids data. It is revealed that the XGBoost (XGBB) algorithm is the most suitable for assessing visibility with high accuracy. It was found that PM_2.5_ and air humidity are of great importance for assessing visibility, while precipitation and wind movement have low indicators when taken into account. Study [35] investigated air pollution in Bishkek (Kyrgyz Republic), which highlights the growing concern due to its impact on public health and the environment. The researchers used ML algorithms and meteorological datasets to analyze the air quality in the city. The performance of several ML algorithms was evaluated, and a prediction model was built and tested using data collected in 2020. The study also assessed the impact of climate change on air pollution in Bishkek, particularly the frequency of days characterized by temperature inversions and air stagnation. The data obtained showed an increase in atmospheric stability and temperature inversions due to climate change.

In our work, the assessment of the morbidity of the population of Almaty and the relationship with atmospheric air pollution using ML methods to identify priority pollutants is considered. An epidemiological analysis of the RD of the population of Almaty was carried out and covered the period from 2017 to 2022, as well as the level of air pollution with suspended substances for the same years. Simulations of measures to reduce the emissions of harmful substances into the atmosphere using ML methods were carried out. For the quantitative reduction in RDs, recommendations were offered to industrial enterprises, road traffic management organizations, thermal power plants, the Department of Environmental Protection, and local executive bodies.

## 2. Materials and Methods

### 2.1. Data on RD and Air Pollutants

During the epidemiological study, data from 2017 to 2022 for the city of Almaty were used with respect to the primary incidence of the RD of ICD (J00-J99) of residents in 3 age groups, as shown in Figure 1a–h: children from 0 to 14 years inclusive, adolescents whose age group ranges from 15 to 17, and adults who are 18 years and older. Information on the incidence of RD according to ICD 10 was provided by the RSE “National Scientific Center for Health Development named after Salidat Kairbekova” (in accordance with Agreement No. 99/23 of 06.03.2023 with the National Engineering Academy of the Republic of Kazakhstan). The data on pollutants were taken from the monthly bulletin on the state of the atmosphere from the Kazhydromet website [36]. According to the monthly bulletin of Kazhydromet on the environment for monitoring atmospheric air quality in Almaty, 14 indicators were determined for the following substances in the city from 2017 to 2022: suspended particles PM_2.5_ and PM_10_, sulfur dioxide SO_2_, carbon monoxide CO, nitrogen dioxide NO_2_, nitrogen oxide NO, phenol, formaldehyde, and heavy metals—cadmium, Cd, lead, Pb, arsenic, As, chromium, Cr, copper, Cu, and nickel, Ni. 

The increase in the incidence of pneumonia in 2020 is due to coronavirus infection among the adult population. 

### 2.2. Model Formation

Air pollution can have a serious impact on human health, including diseases of the respiratory system. Emissions of various pollutants into the atmosphere can lead to serious consequences, increasing the risk of developing various diseases and increasing mortality. To identify the relationship between pollutants and RD, multiple regression models are considered.

In this study, the connection between RD and atmospheric impurities is determined using the random forest method. We consider data on the following atmospheric pollutants as input data: suspended particles, PM_2.5_ (x_1_) and PM_10_ (x_2_), sulfur dioxide SO_2_ (x_3_), carbon monoxide CO (x_4_), nitrogen dioxide NO_2_ (x_5_), nitrogen oxide NO (x_6_), phenol (x_7_), formaldehyde (x_8_), cadmium Cd (x_9_), lead Pb (x_10_), arsenic As (x_11_), chromium Cr (x_12_), copper Cu (x_13_), and nickel Ni (x_14_). As output data, data on the primary incidence of respiratory diseases were considered, such as pneumonia (y_1_); vasomotor and allergic rhinitis (y_2_); chronic rhinitis, pharyngitis, and nasopharyngitis (y_3_); chronic sinusitis (y_4_); chronic diseases of the tonsils and adenoids (y_5_); chronic and unspecified bronchitis and emphysema (y_6_); bronchial asthma (y_7_); and other chronic obstructive pulmonary diseases (y_8_).

### 2.3. Implementation Algorithm 

In this study, the Python programming language’s analytical packages played a crucial role as indispensable tools. Numpy, matplotlib, pandas, seaborn, scikit-learn, and keras [37,38] were harnessed to implement a well-structured algorithm using key steps.

The initial step involved preparing the input data, a process that included transforming it into a dimensionless form for better consistency and compatibility in subsequent analyses. Once the data were ready, thorough cleaning and formatting procedures were carried out to ensure its quality.

Next, the study focused on identifying the relationships between different parameters. This was achieved by calculating correlation coefficients for all columns, allowing for insights into potential multicollinearity issues that could affect the model’s performance.

For constructing regression models, the study opted for the random forest method. 

To assess the models’ effectiveness, the source data were divided into two subsamples: the test set and the training set. 

The study employed essential model evaluation measures, such as the coefficient of determination (R-squared), mean squared error (MSE), and mean absolute error (MAE) estimates. 

In the end, the analysis of the results and conclusions were considered.

By skillfully implementing this algorithm and leveraging the capabilities of Python’s analytical packages, the study aims to gain a comprehensive understanding of the data’s underlying relationships and develops effective regression models for predictive analysis, facilitating valuable insights for future applications.

## 3. Results

### 3.1. Preliminary Data Analysis

Since the beginning of March 2020, the whole world has witnessed the spread of COVID-19, which has led to serious consequences for human health. One of the most characteristic manifestations of this infection is RDs, which occur in infected individuals.

For a thorough study of the effect of COVID-19 on the spread of RD, an analysis matrix was carried out. To construct the matrix, data on RD for the 2020–2022 period and data on COVID-19 infection for the 2020–2022 period in Almaty were used [39]. The aim of this study was to identify the correlation between COVID-19 infection and the development of RD in infected patients. The analysis matrix provided an opportunity to assess the degree of connection between these two factors (Figure 2).

During the analysis, specific data were obtained that allowed us to draw conclusions about the relationship between the incidence of COVID-19 and RD. The study revealed a negative association between COVID-19 infection and the development of RD. This is because the symptoms of COVID-19 are similar to those of RD. Many infected patients with COVID-19 had pneumonia. Negative correlation indicators between these two factors indicate that the presence of the COVID-19 virus affects the development of RD in infected patients. As a result, in the period from 2020 to 2022, there was a decrease in the rates of RD compared to previous years.

### 3.2. Preliminary Data Analysis

A preliminary correlation analysis of all input and output data was carried out for further consideration, and the data were used as input parameters for the learning algorithm. To determine a closer connection between diseases and harmful emissions, a correlation matrix was constructed using input and output data (Figure 3).

According to the correlation analysis, various lung diseases have a close connection with various harmful substances. Based on the conducted research, the correlation coefficients between RD and pollutants were presented, which showed the greatest impact on morbidity. Multiple regression models were constructed based on the most influential pollutants identified. These multiple regression models helped researchers establish a connection between pollutants and RD, taking into account the influence of other factors. The analysis of such models makes it possible to determine which pollutants have the greatest impact on the incidence of RD, and the degree of their influence could be assessed.

The presented multiple regression models can be useful for taking measures to improve environmental quality and for preventing RD by reducing exposure to the most influential pollutants.

Figure 4 shows the correlation coefficients of RD with pollutants, which have the highest indicators.

Based on the information presented in Figure 4, the next chapter discusses the construction of multiple regression models for each respiratory disease using data on pollutants. In this process, the random forest method was used to create models.

This approach will allow researchers to gain a more complete understanding of the pollutants’ impact on various respiratory diseases, and it will also allow the identification of the most significant factors affecting diseases. The constructed multiple regression models will have the potential to be used in further research and the development of measures in order to reduce the risk of diseases due to environmental pollution.

### 3.3. Metric Estimates of Models

Based on the results of the model’s evaluations (Table 1), it can be concluded that the obtained models have reliable indicators with respect to test data values (R^2^) given the fact that the formation of diseases is influenced not only by pollution in the atmosphere but also by a number of other factors. Also, the root mean square (RMSE) and mean absolute (MAE) errors have acceptable indicators for all constructed models (Figure 5a,b).

## 4. Discussion

The following models have been obtained, which determine the priority pollutants that have the greatest impact on each disease:(1)$$\begin{array}{l} y_{1}={0.26x}_{1}+{0.17x}_{2}+{0.13x}_{3}+0.11x_{4}+0.1x_{5}+0.1x_{6}+0.05x_{7}+0.05x_{8},\\ y_{2}={0.32x}_{2}+{0.41x}_{4}+{0.19x}_{5}+0.08x_{12},\\ y_{3}={0.27x}_{2}+0.11x_{3}+0.22x_{4}+0.15x_{5}+0.12x_{6}+{0.07x}_{7}+{0.06x}_{8},\\ y_{4}={0.28x}_{2}+0.09x_{3}+0.27x_{4}+{0.13x}_{5}+{0.07x}_{7}+{0.09x}_{8}+0.05x_{12},\\ y_{5}={0.31x}_{2}+0.37x_{4}+0.32x_{5},\\ y_{6}=0.28x_{2}+{0.09x}_{3}+0.26x_{4}+0.15x_{5}+{0.07x}_{7}+0.07x_{8}+0.07x_{12},\\ y_{7}=0.27x_{2}+{0.09x}_{3}+0.29x_{4}+{0.18x}_{5}+{0.06x}_{7}+0.07x_{8}+0.06x_{12},\\ y_{8}=0.31x_{2}+0.09x_{3}+0.26x_{4}+0.12x_{5}+0.09x_{7}+0.09x_{8}+0.05x_{12}. \end{array}$$

These models allow us to study the influence of various pollutants on the development of all studied types of lung diseases. Particulate matter, PM_2.5_ (26%) and PM_10_ (17%), and sulfur dioxide (13%) have the greatest influence on pneumonia (y_1_). The total weight of their influence is 56%.

In the case of vasomotor and allergic rhinitis (y_2_), PM_10_ (32%) and carbon monoxide (41%) have the greatest effect (total 73%). 

For chronic diseases of rhinitis, pharyngitis, and nasopharyngitis (y_3_), the formation of the disease is influenced by particulate matters PM_10_ (27%), carbon monoxide (22%), nitrogen dioxide (15%), and nitrogen oxide (12%), and their combined effect is 76%.

In chronic sinusitis (y_4_), PM_10_ (28%), carbon monoxide (27%), and nitrogen dioxide (13%) have the greatest effect, collectively constituting 68%. 

For chronic diseases of the tonsils and adenoids (y_5_), carbon monoxide has the greatest effect (37%). 

In chronic and unspecified bronchitis, emphysema (y_6_), PM_10_ (28%), carbon monoxide (26%), and nitrogen dioxide (15%) have the greatest effect, and their combined effect is 69%. 

For bronchial asthma (y_7_), PM_10_ (27%), carbon monoxide (29%), and nitrogen dioxide (18%) have the greatest impact, together making up 74%. 

In the case of other chronic obstructive pulmonary diseases (y_8_), particles PM_10_ (31%) and carbon monoxide (26%) have the greatest influence on the formation of the disease, and the total weight of their influence is 57%.

The proposed method of constructing the model is part of the platform of a unified ecosystem for collecting and processing atmospheric air-monitoring data in industrial cities. This complex work is carried out within the framework of the program “Development of geoinformation systems and monitoring of environmental objects”.

The result of the work presented in this article is to obtain predictive data on the number of respiratory diseases from the level of pollutants.

The new developments proposed by us, using the machine learning method, taking into account heterogeneous factors, as well as affecting the quality of atmospheric air, are promising in the field of healthcare. A deep machine learning method has been developed that will work in real time with input data from automated monitoring systems (AMS). In the process of increasing the input data, the mathematical model (1) becomes dynamic.

The results of the study of air quality in industrial cities carried out by foreign and domestic scientists indicate a lack of information about atmospheric pollution and note the need for detailed studies [40]. In [41,42], the relationship between air pollution and respiratory diseases is investigated using correlation analysis. The results of the study revealed an increasing linear relationship between PM_10_ and SO_2_ levels and the number of hospitalizations of patients with COPD.

And our studies have revealed a linear relationship between the indicators of PM_10_ and CO pollutants and the number of respiratory diseases. A distinctive feature of our proposed approach from the above methods is the construction of a predictive model using machine learning.

This scientific work will make a significant contribution to the development of GIS, big data analysis tools, and expert systems using machine learning algorithms in ecology. To inform the population, using the results of this work, a special website of the situational center for monitoring and forecasting atmospheric air pollution of an industrial city will be created on the platform of a single ecosystem, and additionally LED screens installed on the streets of the city will be used for operational round–the-clock information about the state of atmospheric air.

The models constructed by the authors of this study were used for the predictive modeling of the above scenarios. According to study [43], substances such as PM_2.5_, PM_10_, NO_x_, and SO_2_ are released into the atmosphere during the active operation of the CHP, and their corresponding proportions in the atmosphere have been empirically determined. The proportions of pollutants from motor transport were also determined empirically according to the methodology specified in [44]. In [44,45], the types of substances that are released by motor vehicles (PM_2.5_, PM_10_, NO_x_, SO_2_, and CO) and their percentages in the polluted airspace of the city, along with other sources of pollution, are determined. 

According to the first scenario with respect to the reduction in pollutants from the CHP, the number of respiratory diseases is reduced by 42–44%. The implementation of the second scenario will lead to a reduction in harmful impurities in the atmosphere, and the number of diseases is reduced by 50–52%. The third scenario associated with an increase in the network and types of public transport is also effective. In this case, the indicators of respiratory diseases will be reduced by 39–41%. Therefore, it is necessary to take measures to improve the health of the population.

According to the environmental bulletin of Kazhydromet, monitoring the air quality in Almaty city showed that the concentrations of heavy metals and other pollutants did not exceed the permissible limits. This can be explained by the fact that Almaty is not an industrial city. There are no major industrial facilities within the city’s boundaries that could release such substances into the city’s air basin. The presence of these substances may be attributed to the large number of motor vehicles, waste processing, and trash incineration.

### Recommended Measures to Reduce the Level of Atmospheric Air Pollution

Using the constructed models, the possibilities of carrying out various measures to reduce emissions are considered. The conducted studies with various possible input data scenarios revealed close relationships between diseases and the emission of pollutants into the atmosphere. The results of the study are qualitatively consistent with well-known works [46,47].

In this regard, the most acceptable recommendations are proposed to industrial enterprises and local executive bodies in order to solve the problem of pollutant emission into the atmosphere, which will significantly reduce the quantitative indicators of respiratory-type morbidity among the population. The introduction of recommendations will reduce air pollution and solve problems of this kind. The Republic of Kazakhstan has extensive capabilities and technological solutions that can limit CO_2_ emissions in the energy and transport sectors, which will contribute to economic development without compromising energy consumption requirements and help reduce the incidence of various ailments. A notable example is the favorable conditions in Kazakhstan for the implementation of a program to reduce CO_2_ emissions in the energy sector, especially in the production of electric and thermal energy via the combustion of natural gas. Several viable solutions that can be implemented quickly include the following:The conversion of boiler plants currently using natural gas for the production of both electric and thermal energy by installing mini combined-heat power plants (CHP).Government organizations and institutions can make decisions on the purchase of electricity at premium prices, similarly to the purchase of electricity from renewable sources. Administrative and economic cooperation would accelerate the introduction of this technology. Such measures can lead to a reduction in CO_2_ emissions by more than 200,000 tons per year in the city of Almaty.Stimulating the transition of thermal power plants based on steam turbine units operating by means of the combustion of natural gas to a combined mode of heat and electricity generation. This method has the potential to reduce annual CO_2_ emissions by about a million tons. Two thermal power plants in Uralsk and Aktobe are already operating on the basis of this approach.The introduction of turbo-expander structures that are similar to hydraulic turbines in the nodes of the gas network, where pressure transitions from high to low occur in the gas pipeline.Promoting the decentralized production of thermal energy by creating local networks with minimal losses.The introduction of hybrid-type public transport, which includes hybrid passenger cars, offers a promising solution to achieve a threefold reduction in fuel consumption. By taking this measure, the city of Almaty can significantly reduce CO_2_ emissions, potentially reaching hundreds of tons per day. By implementing these recommendations via a collaborative effort involving industrial enterprises, thermal power plants (TPPs), transportation management organizations, and administrative institutions, significant progress can be achieved in reducing the emission of fine particulate matter into the atmosphere. Consequently, this would have a positive impact on respiratory health outcomes.

## 5. Conclusions

In this study, the negative impact of urban air pollution on public health was assessed using the ML method. The results of the study confirm that air pollution has significant negative consequences for human health. The correlation analysis showed a small effect of COVID-19 infection on the development of RD. Using the ML method, particularly the random forest method, a model that can predict the level of exposure to air pollution relative to public health was built. This makes it possible to conduct a more accurate and objective risk assessment, and organizations can act appropriately to protect public health. One of the main conclusions of this study is that air pollution, especially with high concentrations of PM_2.5_ and other harmful substances, significantly affects the health of the population. This can lead to the deterioration of the respiratory system, cardiovascular diseases, allergic reactions, and other health problems. The use of the ML method makes it possible to identify complex statistical relationships between air pollution and public health, and it can predict potential risks and consequences, which in turn makes it possible to take targeted measures to reduce air pollution and protect public health. However, it should be taken into account that assessing the negative impact of air pollution on public health is a difficult task, and the ML method is one of the tools to solve it.

In general, this study confirms the need to take measures to reduce air pollution and protect public health. The use of ML methods makes it possible to more accurately and effectively assess the health risks of air pollution, identify the main factors and sources of pollution, and predict potential risks. This opens new opportunities for the development and implementation of measures that can improve air quality and protect public health. One of the advantages of the ML method is its ability to work with a large amount of data, and it can identify complex relationships that may go unnoticed when using traditional statistical methods. This helps obtain more accurate and reliable results that can serve as a basis for the development of effective strategies and measures to reduce air pollution. However, it is important to take into account contextual and local peculiarities since air pollution and its impact on public health may vary in different regions. Thus, the use of machine learning in assessing the negative impact of air pollution on public health is an important step in understanding and combating this problem. This makes it possible to obtain more accurate and objective data that can be used to develop and implement effective measures to reduce air pollution and improve people’s quality of life.

## Figures and Tables

**Figure 1 ijerph-20-06770-f001:**
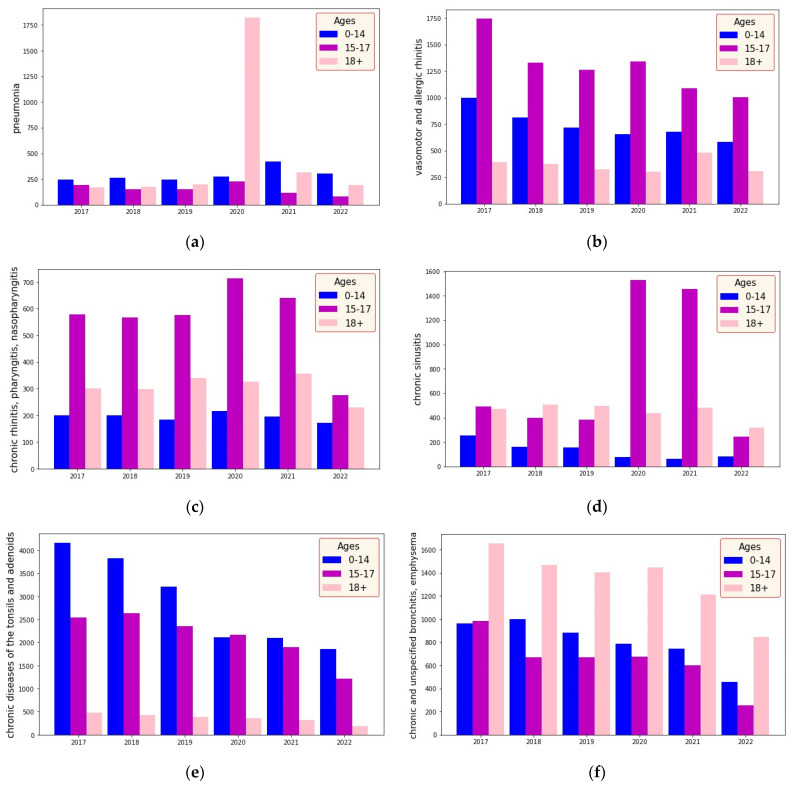
Data on the RD (pneumonia (**a**); vasomotor and allergic rhinitis (**b**); chronic rhinitis, pharyngitis, and nasopharyngitis (**c**); chronic sinusitis (**d**); chronic diseases of the tonsils and adenoids (**e**); chronic and unspecified bronchitis and emphysema (**f)**; bronchial asthma (**g**); and other chronic obstructive pulmonary diseases (**h**)) of the age group from 0 to 14 years, 15–17 years, 18 years and older; the indicators of the spread of diseases per 100 thousand people for all age categories are considered.

**Figure 2 ijerph-20-06770-f002:**
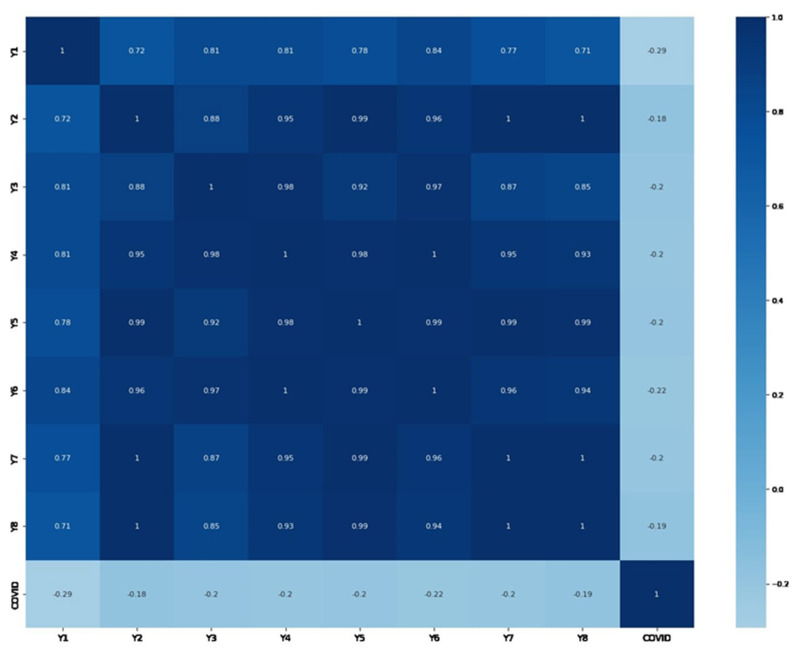
Analysis matrix of correlation between COVID-19 and the development of RD.

**Figure 3 ijerph-20-06770-f003:**
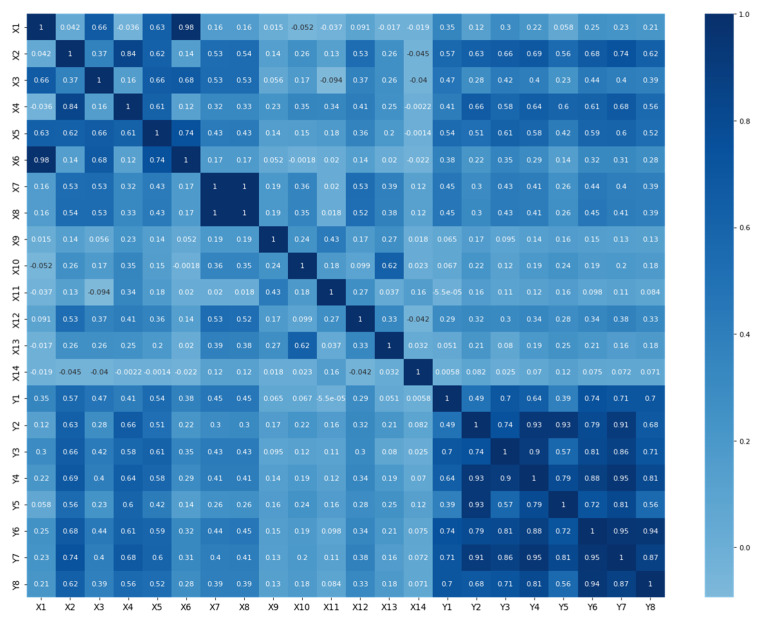
Correlation matrix of indicators of diseases (y_1_-y_8_) and pollutants (x_1_-x_14_).

**Figure 4 ijerph-20-06770-f004:**
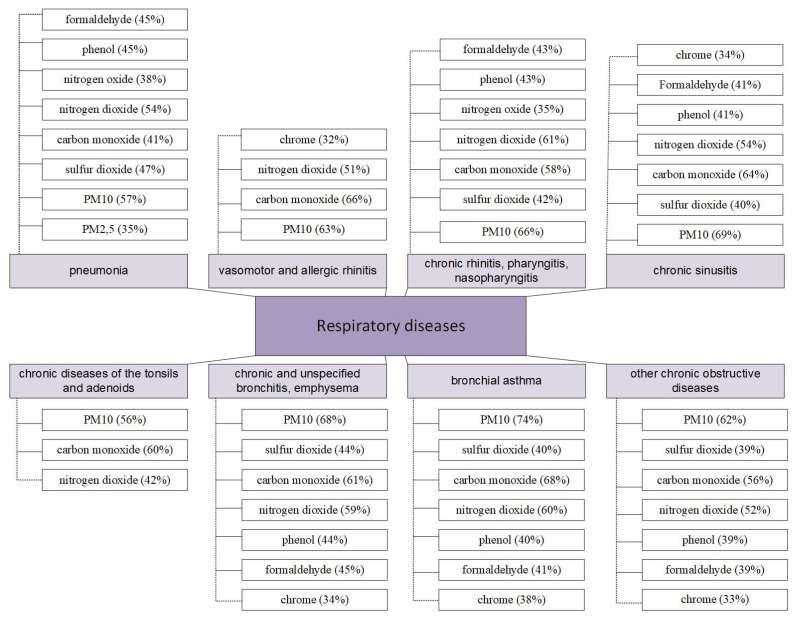
Correlation coefficients of RD with priority pollutants.

**Figure 5 ijerph-20-06770-f005:**
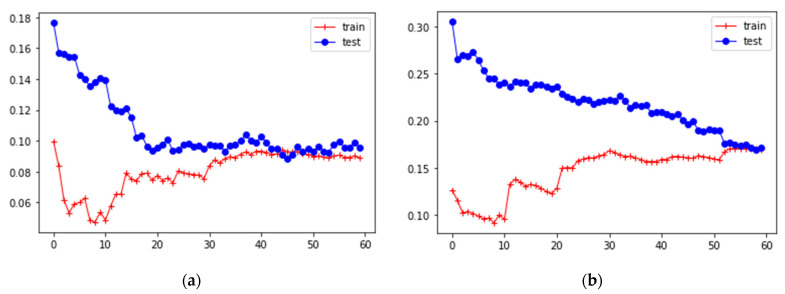
Error curves for training and test data for y_1_ (**a**) and y_2_ (**b**).

**Table 1 ijerph-20-06770-t001:** Metric estimates of models: coefficient of determination (R^2^), mean absolute error (MAE), and standard error (RMSE).

Metrics	Y_1_	Y_2_	Y_3_	Y_4_	Y_5_	Y_6_	Y_7_	Y_8_
R^2^	75%	76%	80%	76%	70%	82%	86%	62%
MAE (per 100 thousand people)	17.6	35.2	21.9	14.4	99.2	42.9	17.0	25.1
RMSE (per 100 thousand people)	34.6	55.9	37.09	25.6	156.8	67.9	35.08	38.5

## Data Availability

Data on the incidence of respiratory diseases of ICD 10 were provided by the RSE “Salidat Kairbekova National Scientific Center for Health Development” (according to Agreement No. 99/23 of 06.03.2023 with the National Engineering Academy of the Republic of Kazakhstan). Data on pollutants were obtained from the monthly bulletin on the state of the environment of Kazhydromet. Available online: https://www.kazhydromet.kz/ru/ecology/ezhemesyachnyy-informacionnyy-byulleten-o-sostoyanii-okruzhayuschey-sredy (accessed on 23 June 2023).

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
