# Peer review of "Assessment of the Negative Impact of Urban Air Pollution on Population Health Using Machine Learning Method"

_ijerph, 2023, doi:10.3390/ijerph20186770_

Round 1

Reviewer 1 Report (Previous Reviewer 1)

Interesting article that investigates air pollution impact on population health using machine learning.

My comments/questions appear by page numbering:

1. page 1, line 36: you mentioned WHO 2017 [1]; however reference [1] is not a WHO refernce

2. page 2, line 53: delete Liu M.B. et al. It is not supposed to be here

3. page 2, line 89: you mentioned various methods were used to predict PM2.5 levels. A very recent study was published by Harvard university is worth mentioning here, include in the reference:

Li, J.,  Kang, CM., Wolfson, J., Alahmad, B., Al-Hemoud, A., Garshick, E., Koutrakis, P. Estimation of fine particulate matter in an arid area from visibility based on machine learning. Journal of Exposure Science & Environmental Epidemiology. 2022: 32, 926-931 

4. page 4, 201: why did you select these specific heavy metals (Cd, Pb, As, Cr, Cu, Ni)? I mean Pm2.5 and PM10 may contain various proportions of these heavy metals (if you do source apportionment). On what basis did you select these six heavy metals?

5. page 6, subsection 2.3. Implementation algorithm: put is a sentence format, not as numbers.

6. page 6, line 254-255: emphasis that this correlation analysis is for Almaty, Kazakhstan

7. page 9, lines 295-297: elaborate on the findings of RF method and the association between RD and pollutants

8. page 9, line 305: I think you mean Figure 5, not Figure 4

page 11, lines 342-343: how about the six heavy metals, how do they contribute to the overall pollution in Almaty?

9. page 13, Reference no. 5: author names should be mentioned first.

Reference 29: include journal name, page numbers

English is fine, minor editing is required.

Author Response

Dear reviewer,
the authors have given answers to all your comments regarding the article. Thank you for your reply.

Sincerely, the authors of the article.

Reviewer 2 Report (Previous Reviewer 2)

This updated manuscript addresses many of my previous comments, however i have the following concerns about the updated manuscript

1. The introduction - this has been changed and now has a largely irrelevant paragraph on what Covid-19 is - it has been added to support the work on Covid and the comments further through - but it is not needed in that detail and seems to be a summary of Covid rather than a few key sentences on Covid. (Why does it matter about vaccination) and it is written in the wrong present tense.

2. The discussion - the first part of teh discussion is basically an extension of the results - with no discussion around any of the sentances

3. The conclusion is improved and helpful

There still needs further English modification

Author Response

Dear reviewer,
the authors have given answers to all your comments regarding the article. Thank you for your reply.

Sincerely, the authors of the article.

Reviewer 3 Report (Previous Reviewer 3)

Overall, the innovation of this article is insufficient, but the workload of empirical analysis masks this drawback. I hope the author can make more innovative research based on this in the future.

Author Response

Dear reviewer,
the authors have given answers to all your comments regarding the article. Thank you for your reply.

Sincerely, the authors of the article.

This manuscript is a resubmission of an earlier submission. The following is a list of the peer review reports and author responses from that submission.

Round 1

Reviewer 1 Report

The article does not show the use of artificial intelligence (AI)as stated in the title of the manuscript. Table 1 presents metric estimates from a statistical multiple regression model. The discussion presents regression equations used in statistics books and does not show where the AI is involved. The recommendation section is not related to the results or discussion. The article is not accepted in its present format.  The article has methodology flaws.

  1. page 1 and 2, lines 40 to 56: sentence is not clear and not understandable of what you are trying to convey. Please re-write the sentence
  2. page 5, Figure 1: the y-axis are not clear, increase the font size
  3. page 6, Figure 2: the map is not clear, location of monitoring points are not legible
  4.  page 7, lines 243 to 260: please summarize the points (1 to 5) in a sentence
  5. page 9, Table 1: are these metric estimates from the statistical regression model? How did you apply artificial intelligence (AI)? what is the output from AI?
  6.  page 10, the equation shown are simple regression equations, how can they add to the strength of your study? How do they relate to AI output?
  7. page 11, Recommended measure: I found that the whole write-up of the recommended measures to reduce the level of atmospheric air pollution is not related to your study. Please be more specific. Talk about the AI model and how it helped prioritize air pollution concentration findings
  8. page 12, line 440: you mentioned that you applied random forest (RF) model: I haven't seen any reference to RF in your manuscript. please clarify
  9.  page 12, line 443: you mentioned 'employing metrics such as coefficient of determination, average absolute error, and quadratic error'. These metrics can be done by regular statistical regression. How can AI helped reach these metrics? what is the value added of the RF model that you used?

The strength of the article is that it claims it used RF AI model to assess urban air pollution in Almaty, however I haven't seen any write-up that relates to this.

The weakness of the article is that it does not show how it employed RF AI to assess the negative impact of air pollution. The title of the manuscript mentions AI and also population health; I haven't seen any reference to the assessment of health impact of the population. Although I would consider rejecting the manuscript, I can recommend reject and resubmit after proper presentation of RF AI assessment of air pollution and association with public health outcomes.

English requires moderate editing. 

Author Response

Please find the response attached

Reviewer 2 Report

Major Comments

- the paragraph reviewing the Liu et al sysematic review is over detailed and includes detail on the statistics that I am unclear as to how they benefit this introduction - it would be preferable to use the introduction as a summary of previous data

- the grammar in the methods section is difficult to understand and appears to be written as if reviewing what was done rather than describe what was done. 

- i am unclear why this paragraph is included 

In the Python programming language, many libraries and tools allow you to work with data more efficiently and conveniently. Among the most common libraries for data processing and analysis are Numpy, Matplotlib, pandas, Seaborn, Scikit-learn and Keras [31]. The Numpy library provides many mathematical functions and a user-friendly in­terface for working with arrays, which makes it an irreplaceable tool for performing ma­trix operations. Matplotlib and Seaborn provide many tools for creating beautiful and in­formative graphs, which makes them essential for data exploration and visualization. The Pandas library provides a userfriendly interface for working with tabular data, and Scikit­learn and Keras provide powerful algorithms for deep learning, which makes them nec­essary for creating accurate and reliable forecasting models [32]. Next, we will describe the random forest (RF) method in more detail.

- random forest method - section 3.1 - this reads like a textbook about what a random forest method is - rather than explaining that it was used— this should be supplementary data

- preliminary data analysis - this is too confusing to read - please just highlight the significant findings

- models - please describe the results - i am unclear what relevance this has

discussion and conclusion

- the discussion does add some insight in to the results - but there needs to be more information in the actual results section. 

How do these fit in to what has previously been done, and why is are these results novel

The grammar overall does not read easily, switching from first person to third person incorrectly.

Author Response

Please find the response attached

Reviewer 3 Report

The main content of the article is an assessment of the negative impact of urban air pollution on population health using artificial intelligence methods. The study analyzes the effects of various air pollutants, with a focus on suspended PM10 particles and carbon monoxide, on respiratory system diseases. The authors utilize the random forest method, a machine learning approach, to uncover hidden patterns between respiratory diseases and all air pollutants. The study concludes that it is imperative to implement measures aimed at reducing air pollution and enhancing overall air quality to prevent the development of chronic respiratory diseases.

I noticed that the article uses relevant data from 2017 to 2022. A big problem is that the author has not discussed and considered the impact of the COVID-19 epidemic on respiratory diseases since 2020. This global infectious disease poses a huge threat to people's respiratory health. No influencing factors and data of the author's inclusion of COVID-19 epidemic were found in either the introduction part or the model method part. This is unscientific and the resulting results are also untrue.

If the author wants to improve the scientificity of the article, he must add an in-depth discussion on the impact of the COVID-19 epidemic on respiratory diseases in the introduction section. In the model part, the influencing factors of COVID-19 epidemic are added.

Author Response

Please find the response attached
